# Type XVIII Collagen Modulates Keratohyalin Granule Formation and Keratinization in Oral Mucosa

**DOI:** 10.3390/ijms20194739

**Published:** 2019-09-24

**Authors:** Ha Thi Thu Nguyen, Mitsuaki Ono, Emilio Satoshi Hara, Taishi Komori, Midori Edamatsu, Tomoko Yonezawa, Aya Kimura-Ono, Kenji Maekawa, Takuo Kuboki, Toshitaka Oohashi

**Affiliations:** 1Department of Molecular Biology and Biochemistry, Okayama University Graduate School of Medicine, Dentistry and Pharmaceutical Sciences, Okayama 700-8558, Japan; thuharhm@gmail.com (H.T.T.N.); m.edamatsu@okayama-u.ac.jp (M.E.); tomoy@cc.okayama-u.ac.jp (T.Y.); oohashi@cc.okayama-u.ac.jp (T.O.); 2Department of Oral Rehabilitation and Regenerative Medicine, Okayama University Graduate School of Medicine, Dentistry and Pharmaceutical Sciences, Okayama 700-8558, Japan; de19016@s.okayama-u.ac.jp (T.K.); a-kimura@md.okayama-u.ac.jp (A.K.-O.); maekawa@md.okayama-u.ac.jp (K.M.); kuboki@md.okayama-u.ac.jp (T.K.); 3Department of Biomaterials, Okayama University Graduate School of Medicine, Dentistry and Pharmaceutical Sciences, Okayama 700-8558, Japan; gmd421209@s.okayama-u.ac.jp

**Keywords:** oral epithelial keratinization, basement membrane, type XVIII collagen

## Abstract

Epithelial keratinization involves complex cellular modifications that provide protection against pathogens and chemical and mechanical injuries. In the oral cavity, keratinized mucosa is also crucial to maintain healthy periodontal or peri-implant tissues. In this study, we investigated the roles of type XVIII collagen, a collagen-glycosaminoglycan featuring an extracellular matrix component present in the basement membrane, in oral mucosal keratinization. Histological analysis of keratinized and non-keratinized oral mucosa showed that type XVIII collagen was highly expressed in keratinized mucosa. Additionally, a 3D culture system using human squamous carcinoma cells (TR146) was used to evaluate and correlate the changes in the expression of type XVIII collagen gene, *COL18A1*, and epithelial keratinization-related markers, e.g., keratin 1 (*KRT1*) and 10 (*KRT10*). The results showed that the increase in *COL18A1* expression followed the increase in *KRT1* and *KRT10* mRNA levels. Additionally, loss-of-function analyses using silencing RNA targeting *COL18A1* mRNA and a *Col18*-knockout (KO) mouse revealed that the absence of type XVIII collagen induces a dramatic decrease in KRT10 expression as well as in the number and size of keratohyalin granules. Together, the results of this study demonstrate the importance of type XVIII collagen in oral mucosal keratinization.

## 1. Introduction

In the oral cavity, the mucous membrane lining the inside of the mouth (oral mucosa) has not only protective and absorptive functions but also assists the functional activities of the oral cavity, such as speaking, chewing, and swallowing. Despite its continuity, the oral mucosa shows specific structures and features to perform diverse functions at each particular position inside the oral cavity [1,2]. The oral mucosa can be divided into three main categories based on its function and histological characteristics, namely: lining mucosa (non-keratinized stratified squamous epithelium), masticatory mucosa (keratinized stratified squamous epithelium), and specialized mucosa (specifically in the regions of the taste buds on lingual papillae) [3]. The lining mucosa, or non-keratinized mucosa, refers to the mucosa in the inner side of the cheeks and mouth floor. The masticatory mucosa, also known as keratinized mucosa, is predominantly dominated by the keratinization of oral epithelial cells and is found on the dorsum of the tongue, hard palate, and attached gingiva [4]. In many studies on oral health condition, the effectiveness of dental treatments, or prevention of oral diseases, researchers are highly concerned about the quality of the keratinized mucosa [5,6,7]. Keratinized mucosa without inflammation and with adequate height and width can give the clinicians an optimistic prognosis of the dental treatment, related for instance, to the risk of gingival recession or implant exposure [5,8].

Epithelial keratinization is known to be regulated by direct epithelial cell–to–cell contact, indirect interaction between epithelial cells or epithelial-mesenchymal cells through paracrine activity of growth factors and signaling molecules, and epithelial cell–to–basement membrane interactions [9,10]. The basement membrane (BM) is a thin layer of a specialized extracellular matrix (ECM) in close apposition to cells and has been demonstrated to not only provide mechanical support and divide tissues into compartments, but also decisively influence cellular behavior [11]. Our previous study indicated that the type IV collagen α6 chain, one major BM component, plays essential roles in the keratinization of oral mucosal epithelial cells [10]. Based on these studies, it is highly possible that other BM constituents could also play important roles in oral mucosal epithelial keratinization.

The major molecular constituents of BMs are type IV collagen, laminins, nidogens, and heparan sulfate proteoglycans [11,12,13,14]. Heparan sulfate proteoglycans (HSPGs) are proteoglycans with the common characteristic of containing one or more covalently attached heparan sulfate (HS) chain, which is a type of glycosaminoglycan (GAG) [15,16]. The secreted extracellular matrix HSPGs class (e.g., perlecan, type XVIII collagen) is present in the BM of the epithelium of many organs, such as skin and kidney [15,17]. HSPGs interact with other matrix components to build up the BM structure and provide a matrix for cell migration [12,15,18,19]. For instance, a mutation of the gene encoding perlecan can result in Schwartz-Jampel syndrome or abnormalities of the skeletal muscles [20]. Type XVIII collagen, unlike other conventional collagens, carries heparan sulfate side chains attached covalently to the core protein; thus, it has characteristics of an HSPG [21]. Mutation of type XVIII collagen is known to lead to the autosomal recessive disorder, Knobloch syndrome, characterized by eye abnormalities [22,23]. A well-known domain of type XVIII collagen is endostatin, which has been reported to inhibit angiogenesis and tumor growth [24]. Seppinen et al. reported that type XVIII collagen and endostatin play an important role in cutaneous wound healing, which is one of the processes of epithelial re-differentiation and re-keratinization [25,26]. Nevertheless, although these reports have suggested that type XVIII collagen could be involved in epithelial keratinization, its function still remains unclear.

In this study, we hypothesized that the HSPG, collagen type XVIII could contribute to the regulatory role of the BM on the keratinization of oral mucosa. The aims of this study were to investigate the difference of the distribution of type XVIII collagen between the keratinized and non-keratinized mucosa and to clarify the role of type XVIII collagen in oral mucosa epithelial keratinization by analysis of the effects of *COL18A1* gene down-regulation in vitro using TR146 cells, and *Col18a1* gene deletion in vivo using a *Col18*-knockout (KO) mouse.

## 2. Results

### 2.1. Immunohistochemical Analysis of Type XVIII Collagen in Bms of Oral Mucosa

The expression of type XVIII collagen was investigated by immunohistochemical (IHC) analysis of BMs in palatal (keratinized) and buccal (non-keratinized) mucosa. As shown in Figure 1, the positive signal of type XVIII collagen was observed in BMs of both tissues; however, interestingly, it was strongly expressed in the BM of the keratinized mucosa.

### 2.2. Functional Analysis of COL18A1 Gene In Vitro

Keratin 1 and keratin 10 are well known as the major keratins of the suprabasal epithelial cell with differentiation and keratinization from the proliferative basal cell layer, while involucrin and filaggrin are expressed in the cornified layer of keratinized mucosa. At first, changes in gene expression of the abovementioned keratinization-related genes were analyzed at different stages of TR146 cell keratinization. As shown in Figure 2A,B, the gene expression levels of keratin 1 (*KRT1*) and keratin 10 (*KRT10*) were stably low in first three days but then increased sharply after seven days of culture. Involucrin (*INV*) and filaggrin (*FLG*) showed the same tendency, with a significant increase after seven days (Figure 2C,D). Interestingly, the gene expression levels of *COL18A1* and *HSPG2*, two main HSPG members in the ECM of epithelial tissue, increased after three days, but more dramatically after seven days of culture (Figure 2E,F).

To have a deeper insight into the effect of type XVIII collagen on mucosal keratinization, we analyzed the effect of *COL18A1* silencing by *COL18A1* siRNA transfection on the gene expression of keratinization-related genes. As shown in Figure 3A, the mRNA levels of the *COL18A1* gene were successfully down-regulated after siRNA transfection and could be maintained even after seven days of transfection. Interestingly, the mRNA levels of keratinization-related genes, *KRT1* (60%), *KRT10* (30%), and *INV* (30%) also decreased significantly, except for *FLG* (not significant) (Figure 3B–E), indicating that type XVIII collagen could play an essential role in oral mucosa keratinization.

### 2.3. Histological Analysis of Keratinized Oral Mucosa between Wide-Type and Col18-KO Mice

Next, to analyze the function of type XVIII collagen in oral epithelial keratinization in vivo, we analyzed the phenotype of *Col18*-KO mice. Comparison of the oral epithelial keratinization by hematoxylin and eosin (HE) staining showed no difference between wide type (WT) mice and *Col18*-KO mice (Figure 4A). However, IHC and quantitative analysis of KRT10 expression in the keratinized gingiva from WT and *Col18*-KO mice revealed that the percentage of positive signal area for KRT10 in the oral mucosa epithelium was significantly weaker and sparser and two-fold lower in *Col18*-KO mice than in the WT mice (Figure 4C).

### 2.4. Ultrastructural Analysis of Keratinized Oral Mucosa between Wide-Type and Col18-KO Mice

Finally, TEM analysis was performed to analyze the ultrastructural differences between WT and *Col18*-KO mice. In the prickled layer, the two mice showed similar structural morphology (Figure 5A), but in the granular layer, there was a clear difference regarding the number and size of keratohyalin granules (KHG) (Figure 5B). KHG in the granular layer of WT mice were seen as electron-dense granules with multiform-shape and uniformly distributed, while in the granular layer of *Col18*-KO mice, they were remarkably smaller, thinner, and sparsely distributed. Moreover, further quantitative analyses demonstrated that the number of KHG was not significantly different (Figure 5C), but the mean size of KHG was significantly smaller in *Col18*-KO mice (Figure 5D). Correspondingly, the percentage area of KHG in the total measured area was also clearly lower in *Col18*-KO mice. These data indicated that maturation of KHG was inhibited in *Col18*-KO mice and, collectively, these results demonstrated that type XVIII collagen is one of the regulators of oral mucosal keratinization.

## 3. Discussion

Epithelial keratinization is an important barrier against pathogens and mechanical stress. Additionally, the keratinized gingiva in the oral cavity, for instance surrounding a tooth or dental implant, is crucial to maintain healthy periodontal and peri-implant tissues. A width of keratinized mucosa of more than 2 mm is required to maintain the tooth/implant longevity; however, insufficient width of keratinized mucosa (<2 mm) could lead to mucosal recession and attachment loss, and increased risk of periodontal and peri-implant diseases [27,28].

It is well known that BM is deeply involved in determining epithelial cell fate, such as proliferation, differentiation, and maturation [10,29,30]. Indeed, recently, we also demonstrated that the type IV collagen α6 chain regulates the keratinization of oral mucosal epithelial cells [10]. However, the relationship between the keratinization of oral mucosa and other BM constituents is still unclear. Type XVIII collagen belongs to the secreted extracellular matrix class of HSPGs, the same class with perlecan. Type XVIII collagen and perlecan are well-known BM proteins that play well-defined functions in several tissues, including vessels, skin, and kidney. It has been reported that type XVIII collagen and perlecan are involved in age-related epithelial keratinization and wound healing of skin [26,31]. However, their functions had not been thoroughly investigated in oral mucosa.

Our IHC analysis showed that the expression levels of type XVIII collagen and perlecan were significantly different in keratinized mucosa and non-keratinized mucosa (Figure 4 and Appendix A). While a deficiency in *Hspg2*, the perlecan encoding gene, is lethal, *Col18a1* deletion is not, and therefore allowed the analysis of oral mucosal epithelial keratinization in vivo. The results revealed the down-regulation of KRT10 in palatal mucosa and the marked decrease in the number and size of KHG in the granular layer of the epithelium of *Col18*-KO mice. The function of KHG has been indicated as a structure converting keratin tonofilaments into a homogenous keratin matrix, promoting the formation of the epithelial cornified cell envelope, also known as cornification or keratinization [32]. Therefore, these results strongly indicate type XVIII collagen is one important factor that regulates the keratinization of oral epithelial mucosa.

There are several reports on the function of type XVIII collagen both in humans and mice [22,23,25,33]. A well-known syndrome of the mutation in type XVIII collagen is Knobloch syndrome, an autosomal recessively inherited disease, characterized by the occurrence of high myopia, vitreoretinal degeneration with retinal detachment, macular abnormalities, and occipital encephalocele, which is a neural tube closure defect [22,23]. Importantly, the lack of collagen XVIII leads to a lack of antiangiogenic endostatin domain, which is a C-terminal proteolytic fragment of type XVIII collagen and able to inhibit angiogenesis and tumor growth by restricting endothelial cell proliferation and migration [24]. Seppinen et al. reported that cutaneous wound healing was accelerated in *Col18*-KO mice but delayed in endostatin transgenic mice [25,26]. Based on these reports, it can be assumed that not only type XVIII collagen, but also endostatin itself might be related to inhibition of epithelial keratinization of mucosa in *Col18*-KO mice. In addition, type XVIII collagen is known to have three distinct variants, i.e., the short, medium, and long isoforms. The short isoform is transcribed from promoter 1 in exon 1, while the middle and long isoforms are transcribed from promoter 2 in exon 3 [34,35]. These isoforms differ from each other in terms of their N-terminal non-collagenous (NC) terminus, tissue distribution, and functions. Indeed, it has been reported that the short isoform of type XVIII collagen is the dominant form in epithelial and endothelial BMs, and the medium and long isoforms exist in perisinusoidal spaces in the liver and the glomeruli in the kidney [36,37]. Our group also recently reported that only the short isoform of type XVIII collagen increased in the early stages of the wound healing process [38]. Moreover, *Col18a1*^p1/p1^ mice, which refers to the *Col18a1* promoter 1 specific KO mice lacking expression of the short isoform, and *Col18a1*^p2/p2^ mice, which refers to the *Col18a1* promoter 2 specific KO mice lacking expression of the medium and long isoforms, have shown different phenotypes in the kidney [39]. From these reports, it can be estimated that the isoform-specific distribution and functions can also be observed in the keratinization of oral mucosa. Further investigations are necessary to understand more deeply the distribution and function of type XVIII collagen isoforms.

Additionally, perlecan, which is basically composed of five distinct domains including an N-terminal domain containing three attachment sites for HS chains, interacts with numerous growth factors, such as fibroblast growth factor (FGF) and vascular endothelial growth factor (VEGF) [40,41,42]. On the other hand, perlecan is the most abundant proteoglycan and contributes to the complex structure of BMs via its binding properties to other BM components, including type IV collagen, laminin, and nidogen [18,43,44], and its anchoring role at BM-connective tissue junctions by the high affinity interaction of proline arginine-rich end leucine repeat protein (PRELP) and the HS chains [45]. In addition, other HSPGs in the ECM of epithelial tissue, such as type XVIII collagen and agrin, which also contain HS chains in their structure, may also presumably interact with PRELP to form BM anchors [45]. Therefore, HSPGs in general, and more specifically perlecan, are not only essential for embryonic development but also play important roles in the homeostasis of tissues and organs [13,45,46,47,48,49]. Although the number of reports focusing on oral mucosa is limited, there have been numerous studies on skin tissue showing that perlecan is essential for epidermal morphogenesis [50,51] and for the maintenance of the self-renewal capacity of basal keratinocytes [51]. From these reports and our IHC data showing high levels of perlecan in keratinized mucosa (Appendix A), it can be speculated that perlecan might also be an important regulator of the keratinization of oral mucosa. The perlecan function at the BM may not only be due to its own and direct roles in epithelial tissue keratinization, but it may also be due to its interaction with other BM components. In fact, several studies have reported the relationship between type XVIII collagen and perlecan. For instance, Sasaki et al. showed that the C-terminal domain NC1, containing a trimerization domain, a protease-sensitive region, and an endostatin domain, of type XVIII collagen strongly interacts with perlecan in vitro [51], and Sylvie et al. further showed that endostatin binding to HS depends on divalent cations in vitro [52]. Moreover, it has been reported that endostatin domain of type XVIII collagen and perlecan are colocalized in BM in vivo [53]. From these reports, it could be presumed that type XVIII collagen and perlecan could bind to each other, and therefore, a compensation from other HSPG members could have provided HS to partly assume the vacant role of type XVIII collagen in *Col18*-KO mice [41].

Another basement membrane component, type IV collagen α6, has also been proved to play important roles in keratinization of oral mucosa [10]. In comparison with *Col4a6*, *Col18*-KO mice showed no difference in keratinization of oral mucosa at the newborn stage (data not shown) but showed a clear difference at adulthood (Figure 4B,C). In contrast, a suppressed keratinization was observed in both newborn and adult *Col4a6*-KO mice. At each stage of development, the functions and interactions of proteins are diverse and, therefore, both type XVIII collagen and the type IV collagen α6 chain might be important for the maintenance of homeostasis, but only the type IV collagen α6 chain would be important for the initial development of keratinized oral mucosa.

In summary, our study demonstrated that down-regulation of *COL18A1* in vitro resulted in inhibition of keratinization of epithelial cells. Moreover, deletion of *Col18a1* in vivo led to an inadequate keratinization phenotype of oral mucosa in the *Col18*-KO mice, as demonstrated by a suppressed KRT10 expression and remarkably smaller size of KHG in the granular layer of epithelial tissue in those mice. These results indicate that type XVIII collagen is a modulator of oral epithelial mucosal keratinization.

## 4. Materials and Methods

### 4.1. Cells and Culture Methods

Human squamous carcinoma cells (TR146) purchased from DS Pharma Biomedical Co., Ltd. (Osaka, Japan) were used as human oral mucosa-derived epithelial cells. TR146 cells were cultured in F-12 Nutrient Mixture (Ham’s F-12) medium (Life Technologies, Gaithersburg, MD, USA) containing 10% fetal bovine serum (FBS; Life Technologies, Gaithersburg, MD, USA), 2 mM·L-glutamine (Life Technologies, Gaithersburg, MD, USA), and antibiotics at 37 °C in 5% CO_2_. When the cultures reached sub-confluence, the cells were harvested with Accutase (Innovative Cell Technologies, San Diego, CA, USA), and passaged according to the conventional method. To promote the differentiation of epithelial cells, a 3-dimensional (3D) culture method was carried out using a ThinCert cell culture insert (Greiner Japan, Tokyo, Japan) in a 12 well ThinCert plate, as reported [10]. Briefly, TR146 cells (5 × 10^5^ cells/plate) were seeded in the upper chamber with 1.0 mL of medium. Another 5 mL of medium was added into the lower chamber. After 24 h and confirmation that the TR146 cells had adhered to the cell culture insert, all the medium in the upper chamber was aspirated and the medium in the lower chamber was replaced by new 4 mL medium. The medium in the lower chamber was changed every 2 days. TR146 cells were collected immediately after seeding (0 h) or at 1, 3, and 7 days after seeding for gene expression analysis.

For down-regulation of the *COL18A1* gene, 20 nM of siRNA targeting the *COL18A1* gene (StelthTM Si*COL18A1*; Life Technologies, Gaithersburg, MD, USA) was transfected into TR146 cells using Lipofectamine RNAiMAX (Life Technologies, Gaithersburg, MD, USA), according to the manufacturer’s instruction. StelthTM RNAi Negative Control High GC Duplex (Life Technologies, Gaithersburg, MD, USA) was used as the negative control. After transfection, TR146 cells (2.5 × 10^6^ cells/plate) were cultured in the ThinCert cell culture inserts, as described above, and cultured for 7 days.

### 4.2. Real-Time RT PCR Analysis

Total RNA was collected and purified using the PureLink RNA Mini Kit (Life Technologies, Gaithersburg, MD, USA), following the manufacturer’s instruction. The cDNA reverse-transcription was conducted using the iScript cDNA synthesis kit (Bio-Rad, Hercules, CA, USA) with purified RNA samples. Real-time RT-PCR was performed to quantify the expression of the target gene by using the KAPA SYBR FAST qPCR Master Mix (KAPA BIOSYSTEMS, Wilmington, MA, USA) and CFX96 real-time system (Bio-Rad). The reference gene *S29* was used to normalize the levels of mRNAs of interest. Primer sequences are shown in Table 1.

### 4.3. Animals

Eight–week–old c57BL/6 J mice (wild type, WT) were purchased from CLEA Japan (Tokyo, Japan). *Col18*-KO mice were generated in Dr. Bjorn Olsen’s laboratory at Harvard Medical School [33] and backcrossed with C57BL/6 J for over fifteen generations. The animal experiment protocols used in this study (OKU-2016375, OKU-2017051) were approved by the Okayama University Animal Research Committee. All animals were handled according to the guidelines of the Okayama University Animal Research Committee.

### 4.4. Hematoxylin and EOSIN Staining

Mouse heads without skin were collected, fixed with 4% paraformaldehyde (PFA) for 3 days, and decalcified in Morse Solution (FUJIFILM Wako Pure Chemical Corporation, Osaka, Japan) within one week. Standard hematoxylin and eosin (HE) staining was performed for histological observation.

### 4.5. Immunohistochemical Staining

Our previous histological and morphological analysis showed that the protein expression level of KRT10 and strata corneum and granulosum (granular layer), where KHGs can be observed, are clearly identified in the keratinized palatal mucosa, but not in non-keratinized buccal mucosa of mice [5,10,25]. Therefore, in this study, the palatal and buccal mucosa were used as keratinized and non-keratinized mucosa, respectively, for the comparative analysis of the type XVIII collagen expression level.

Freshly isolated, non-fixed, and un-decalcified tissues were embedded into a super cryoembedding medium (SECTION-LAB Co. Ltd., Hiroshima, Japan) and frozen in hexane cooled by dry ice. Histological sections, 5 µm thick, were made by using a tungsten carbide blade and adhesive cryofilms, according to Kawamoto’s film method, and using a cryostat machine (Leica 3050S, Leica Biosystems Nussloch GmbH, Nussloch, Germany). Samples were fixed in acetone for 20 min at room temperature (RT). Next, sections were blocked in 5% normal goat serum (Life technologies) containing 1% BSA (Sigma, St Louis, MO, USA)  for 1 h at RT and incubated with primary antibody overnight at 4  °C. Monoclonal primary antibody, CM186, against NC1 domain of mouse collagen XVIII was a kind gift from Dr. Sado [54], and anti-KRT10 antibody (ab76318) was purchased from Abcam (Cambridge, UK). Sections were then washed, and incubated with the secondary antibodies, Alexa Fluor 488 donkey anti-rabbit IgG (Life technologies, Gaithersburg, MD, USA) or Alexa Fluor 488 donkey anti-rat IgG (Life technologies, Gaithersburg, MD, USA), for 1 h at RT in a dark chamber. DAPI (Life technologies, Gaithersburg, MD, USA) was used simultaneously to stain the nuclei. All images were taken by a BZ-700 fluorescence microscope (Keyence, Osaka, Japan) and quantification of KRT10 positive area in the epithelium was analyzed with the BZ analyzer (Keyence).

### 4.6. Transmission Electronic Microscope

The palatal mucosa of WT and *Col18*-KO mice were harvested, and all samples were fixed with 2% PFA and 2% glutaraldehyde in phosphate overnight. The samples were prepared according to a previously reported method [10]. Briefly, the samples, post-fixed with 1% osmium tetroxide and dehydrated in ethanol, were embedded using a Spurr Low-Viscosity Embedding Kit (Polysciences, Warrington, PA, US) and cut in ultrathin sections (LEICA EM UC7; Leica Mikro-systeme, Vienna, Austria). The ultrathin sections, stained with uranyl acetate and lead citrate, were observed using a transmission electron microscope (TEM; H-7650, HITACHI, Tokyo, Japan). ImageJ software was used to count the number of, and measure the size and area of keratohyalin granules based on the TEM images.

### 4.7. Statistical Analysis

The results obtained from quantitative experiments were reported as the mean values  ±  SD. Statistical analyses were performed with one-way factorial ANOVA followed by Tukey’s multiple comparison tests or Student’s unpaired *t*-tests, when appropriate. The *p*-value was used in statistical hypothesis testing, and a *p*-value less than 0.05 was considered significant. In the figures, *p*-value levels are described as * *p* ≤ 0.05, ** *p* ≤ 0.01 and *** *p* ≤ 0.001, and NS means non-significant difference.

## Figures and Tables

**Figure 1 ijms-20-04739-f001:**
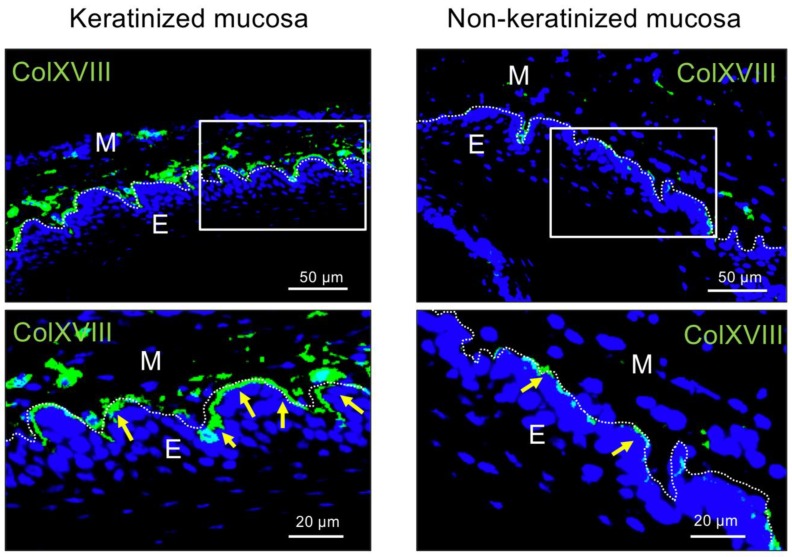
Immunohistochemical staining of type XVIII collagen in the basement membrane (BM) of keratinized mucosa and non-keratinized mucosa. All sections were cut in the coronal direction. Boxes indicate the area shown at higher magnification in the lower panels. Yellow arrows indicate the positive signal of type XVIII collagen. Note that type XVIII collagen (green) is highly expressed in keratinized mucosa. Nuclei were counterstained with DAPI (blue). E, epithelial tissue; M, mesenchymal tissue. Dashed lines indicate the basement membrane. Results are representative of three independent experiments.

**Figure 2 ijms-20-04739-f002:**
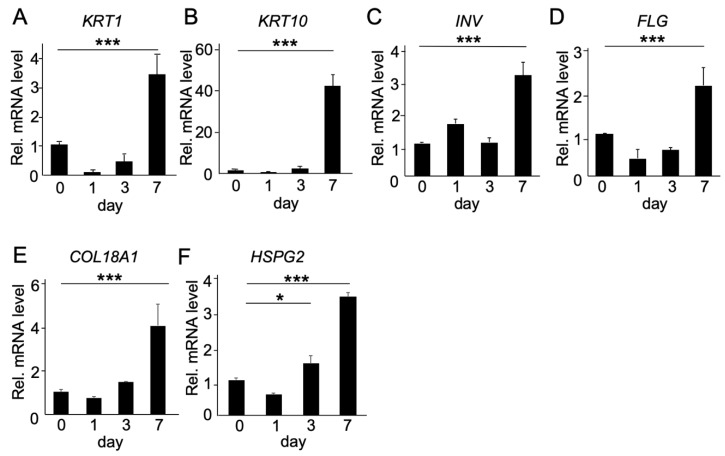
Quantitative analysis of gene expression of keratinization-associated genes, and the heparan sulfate proteoglycans, *COL18A1* and *HSPG2*, during epithelial keratinization in vitro. TR146 cells were cultured onto a ThinCert cell culture insert (3D culture). Samples were collected after each time-point (0, 1, 3, 7 days) and the gene expression levels of *KRT1*, *KRT10*, *INV*, *FLG*, *COL18A1*, and *HSPG2* were analyzed by real-time RT-PCR and normalized with levels of *S29* ribosome RNA. Note that the mRNA expression levels of most genes *KRT1* (**A**), *KRT10* (**B**), *INV* (**C**), *FLG* (**D**), *COL18A1* (**E**), and *HSPG2* (**F**) increased significantly after 7 days of culture. Bars represent the mean values and standard deviation (+/−SD) (*n* = 3). * *p* < 0.05, *** *p* < 0.001 (ANOVA, Turkey multiple comparison test).

**Figure 3 ijms-20-04739-f003:**
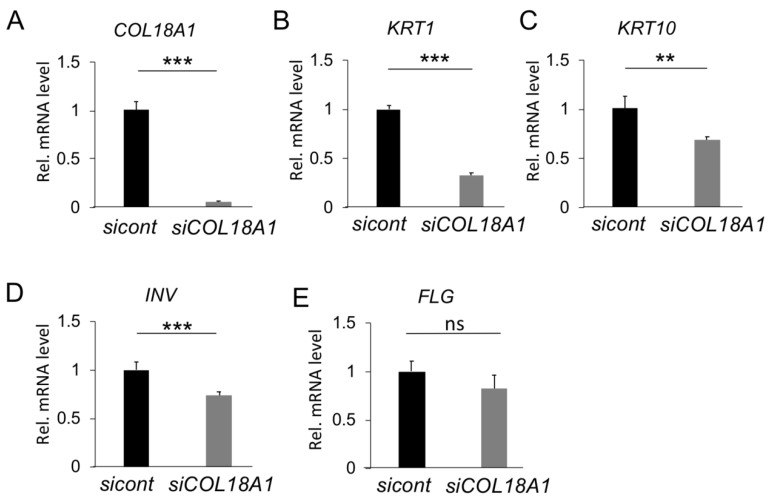
Functional analysis of *COL18A1* knockdown in vitro. SiRNA targeting the COL18A1 gene was transfected into TR146 cells, then transfected cells was subsequently cultured for 7 days in a 3D culture method. The mRNA expression level of *COL18A1* dropped sharply by *COL18A1* siRNA transfection, on the seventh day of culture (**A**). Consequently, the gene expression of *KRT1* and *KRT10* (**B**,**C**) and *INV* (**D**) also decreased remarkably, except for that of *FLG* (**E**). Bars represent the mean values and standard deviation (+/−SD) (*n* = 3). ** *p*  <  0.01, *** *p*  <  0.001, ns: no significant difference (Student’s *t*-tests). Results are representative of at least three independent experiments.

**Figure 4 ijms-20-04739-f004:**
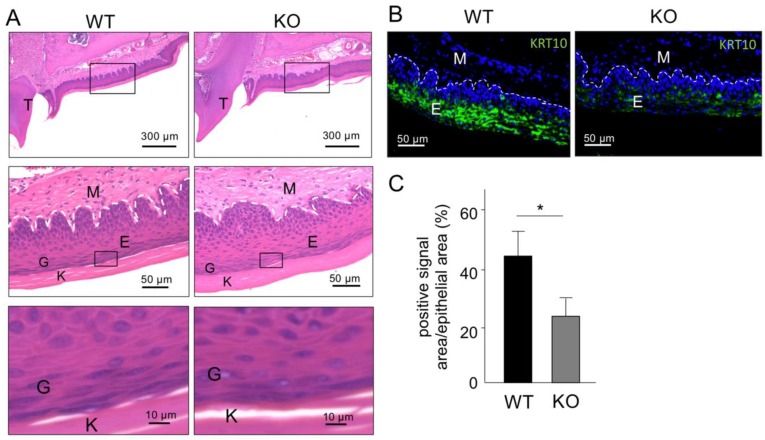
Histological comparison of keratinized mucosa between WT and *Col18*-KO mice. (**A**) HE staining images of palatal mucosa of WT and *Col18*-KO mice. Boxes indicate the area shown at higher magnification in the lower panels. Note that there is no difference between WT and *Col18*-KO mice. (**B**) IHC staining images for KRT10 (green). (**C**) The percentage of positive signal for KRT10 in the area of palatal mucosa. Bars represent the mean values and standard deviation (+/−SD) (*n* = 3). * *p*  <  0.05 (Student’s *t*-tests). Note that the expression levels of KRT10 are lower in *Col18*-KO mice compared to WT mice. T, tooth; G, granular layer; K, keratinized (cornified) layer. Results are representative of three independent experiments.

**Figure 5 ijms-20-04739-f005:**
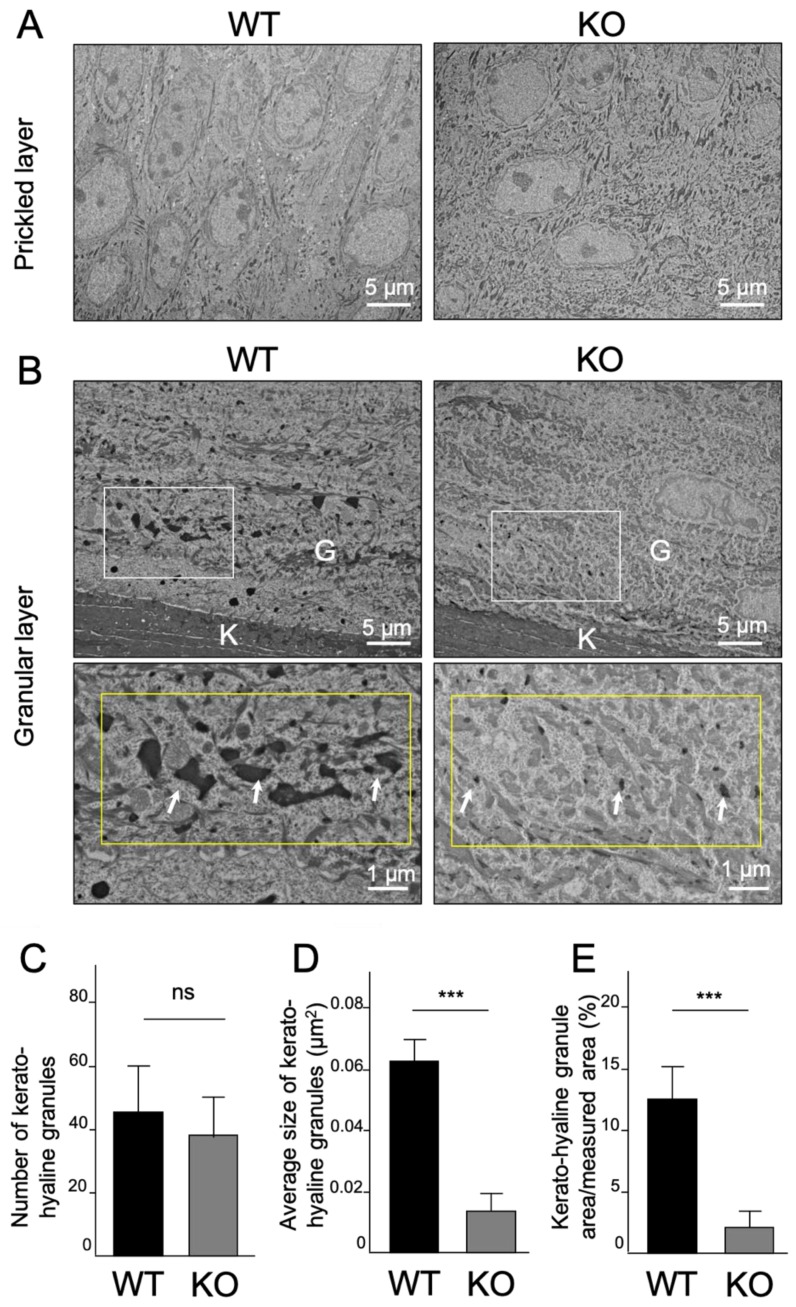
Ultrastructural comparison of keratinized mucosa between WT and *Col18*-KO mice. (**A**,**B**) Transmission electron microscope (TEM) images of (A) prickled layer and (B) granular layer in palatal mucosa of WT and *Col18*-KO mice. Arrows indicate keratohyalin granules (KHG) in the granular layer. White boxes indicate the area shown at higher magnification in the lower panels and yellow boxes indicate the total measured area for calculation of the ratio between the KHG area and total measured area. The number, average size, and percentage area of KHG are shown in graph (**C**), (**D**), and (**E**), respectively. Bars represent the mean values and standard deviation (+/−SD) (*n* = 4). *** *p*  <  0.001, ns: no significant difference (Student’s *t*-tests). Note that a clear difference in the size and area of KHG could be observed in the granular layer, but not in the prickled layer.

**Table 1 ijms-20-04739-t001:** The base sequence of the primers used for RT-PCR. S: sense, AS: antisense.

Target Gene	Encoded Protein	Type	GeneBank Registration Number	Primer Set
*S29*	Ribosome protein S29	Human	BC032813	5′-TCTCGCTCTTGTCGTGTCTGTTC-3′(S)
				5′-ACACTGGCGGCACATATTGAGG-3′(AS)
*COL18A1*	Type XVIII collagen	Human	BC063833	5′-TCCAGAGAATGCCGCTTG-3′(S)
				5′-GGAACTTGTCAGGGTCCG-3′(AS)
*KRT1*	Keratin 1	Human	BC063697	5′-CTTACTCTACCTTGCTCCTACT-3′(S)
				5′-AAATCTCCCACCACCTCC-3′(AS)
*KRT10*	Keratin 10	Human	NM_000421	5′-GCATCACCATGTCTGTTC-3′(S)
				5′-GCTAGAAATTCTTAGGGATGAC-3′(AS)
*INV*	Involucrin	Human	BC046391	5′-CCTCAGATCGTCTCATACAAG-3′(S)
				5′-ACAGAGTCAAGTTCACAGATG -3′(AS)
*FLG*	Filaggrin	Human	NM_002016	5′-AGACTCTAGTACCGCTAAGG-3′(S)
				5′-CGTGACTGTATTCCTGAGTG-3′(AS)
*HSPG2*	Perlecan	Human	M85289	5′-GCCTTCACTTCCAGATGG -3′(S)
				5′-CCACCCCAACTCTTACCA-3′(AS)

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
