# Peer review of "Type XVIII Collagen Modulates Keratohyalin Granule Formation and Keratinization in Oral Mucosa"

_ijms, 2019, doi:10.3390/ijms20194739_

Round 1
Reviewer 1 Report
This is an interesting paper focuses on investigating the roles of type XVIII collagen in oral mucosa keratinization. Introduction provides sufficient background, methodological design is appropriate, and conclusions are supported by interesting results and findings. This is a well-written paper with a significant novelty. However, there are some minor concerns:
Lines 73-79: Introduction section should not be include findings, please remove these sentences and replace with text describing the aims and hypothesis of this study.
Lines 82-86: These sentences describing the rationale of selecting buccal mucosa/palatinum as test tissues should be transfer to Materials and Methods section.
Lines 99-101 As previously, the reasons for selecting 3D TR146 cell culture model should be describe in Materials and Methods section.
Figures 2-4: To enhance clarity of presentation, axis titles and other symbols should be written in larger font size in the figures. Please edit these figures.
Statistical analysis. Different p-value levels are used to indicate the strength of statistical significance in Figures. Please provide this description to the section 4.7. (lines 301-)
Author Response
This is an interesting paper focuses on investigating the roles of type XVIII collagen in oral mucosa keratinization. Introduction provides sufficient background, methodological design is appropriate, and conclusions are supported by interesting results and findings. This is a well-written paper with a significant novelty. However, there are some minor concerns:
Comment #1:Lines 73-79: Introduction section should not be included findings, please remove these sentences and replace with text describing the aims and hypothesis of this study.
Response #1:The authors thank the reviewer’s comment, and have removed the sentences describing the research findings from the introduction section.
Comment #2:Lines 82-86: These sentences describing the rationale of selecting buccal mucosa/palatinum as test tissues should be transfered to Materials and Methods section.
Response #2:The authors thank the reviewer’s comment, and have moved this information to Materials and Methods section.
Comment #3:Lines 99-101 As previously, the reasons for selecting 3D TR146 cell culture model should be describe in Materials and Methods section.
Response #3:The authors thank the reviewer’s comment, and have moved this information to Materials and Methods section.
Comment #4:Figures 2-4: To enhance clarity of presentation, axis titles and other symbols should be written in larger font size in the figures. Please edit these figures.
Response #4:The authors thank the detailed observation made by the reviewer, and have edited the font size in the figures.
Comment #5:Statistical analysis. Different p-value levels are used to indicate the strength of statistical significance in Figures. Please provide this description to the section 4.7. (lines 301-)
Response #5:The authors thank the reviewer’s comment, and have included the description of the p-value levels in the section 4.7.
Reviewer 2 Report
Ijms 591372 Review
This is an extremely short manuscript and is probably the shortest publication I have ever been given to review. The authors don't really discuss much of substance in their discussion which is also a reflection on the paucity of data they have generated. The authors are encouraged to expand on their comments in a revised version of the manuscript and to consult with other MDPI publications to see how long these are and the in-depth commentary they cover.
I would have liked to find out what features of type XVIII collagen and perlecan structure were important for the functional properties of basement membrane but the authors did not cover this or an appraisal of what is known in the literature. The literature review is very poor with only a total of 29 publications cited and many are dated. The authors should provide current knowledge on the subject of their manuscript to be worthy of publication. This is potentially achievable with inclusion and expansion of the additional information I have provided.
General comment –when citing a publication(s) in square brackets make sure they are within the boundaries of a sentence, then close the sentence with a full stop.
Line 58 “Heparan sulfate proteoglycans (HSPGs) are glycoproteins,” - HSPGs are proteoglycans as you have already stated here and not glycoproteins –you need to correct this statement.
All of the figures need some attention.
Figure 1 legend needs to be more explanatory. What is the dotted line signifying, what are the yellow arrows labeling. The fluors should be described and the M and E labels explained. The duplicated comment above the legend is not required. Use the legend to Figure 5 as a guide for the amount of information required in a legend. If the comments on lines 93-97 are supposed to be part of the legend they should follow on continuously with the other information. The same applies for lines 114-120 for Figure 2 and similar features in the other figures.
Figure 2, 3, legends need to be more informative. Explain the genes that were measured and what the error bars represent. What was n in this experiment. Explain the statistical comparisons and provide P values. The duplicated comment above the legend is not required. Use the legend to Figure 5 as a guide for the amount of information required in a legend.
Figure 4 legend needs more information. What stain was used in A. Explain annotations of labels used. The duplicated comment above the legend is not required. Use the legend to Figure 5 as a guide for the amount of information required in a legend.
Legend to Figure 5. The duplicated comment above the legend is not required.
Table 1 should be provided in the correct format for the journal. Descriptive Title required for the Table.
Only 29 references are cited in this study which is very low, the literature does not appear to have been thoroughly examined.
The following relevant references are missing from the literature cited which does not seem to be very extensive. Ref 5 demonstrates a means of anchoring perlecan to the basement membrane which is relevant to the background of this study and to the functional characteristics of this tissue. The other references below provide important background information for this tissue.
1: Dos Santos M, Michopoulou A, André-Frei V, Boulesteix S, Guicher C, Dayan G, Whitelock J, Damour O, Rousselle P. Perlecan expression influences the keratin 15-positive cell population fate in the epidermis of aging skin. Aging (Albany NY). 2016 Apr;8(4):751-68. doi: 10.18632/aging.100928.
2: Sher I, Zisman-Rozen S, Eliahu L, Whitelock JM, Maas-Szabowski N, Yamada Y, Breitkreutz D, Fusenig NE, Arikawa-Hirasawa E, Iozzo RV, Bergman R, Ron D. Targeting perlecan in human keratinocytes reveals novel roles for perlecan in epidermal formation. J Biol Chem. 2006 Feb 24;281(8):5178-87.
3: Ghiselli G, Eichstetter I, Iozzo RV. A role for the perlecan protein core in the activation of the keratinocyte growth factor receptor. Biochem J. 2001 Oct 1;359(Pt 1):153-63.
Knox S, Fosang AJ, Last K, Melrose J, Whitelock J. Perlecan from human epithelial cells is a hybrid heparan/chondroitin/keratan sulfate proteoglycan. FEBS Lett. 2005 Sep 12;579(22):5019-23. Bengtsson E, Mörgelin M, Sasaki T, Timpl R, Heinegård D, Aspberg A. The leucine-rich repeat protein PRELP binds perlecan and collagens and may function as a basement membrane anchor. J Biol Chem. 2002 Apr 26;277(17):15061-8. Gatseva A, Sin YY, Brezzo G, Van Agtmael T. Basement membrane collagens and disease mechanisms. Essays Biochem. 2019 Aug 6. pii: EBC20180071. Sekiguchi R, Yamada KM. Basement Membranes in Development and Disease. Curr Top Dev Biol. 2018;130:143-191.8. Pozzi A, Yurchenco PD, Iozzo RV. The nature and biology of basement membranes. Matrix Biol. 2017 Jan;57-58:1-11.
Halfter W, Oertle P, Monnier CA, Camenzind L, Reyes-Lua M, Hu H, Candiello J, Labilloy A, Balasubramani M, Henrich PB, Plodinec M. New concepts in basement membrane biology. FEBS J. 2015 Dec;282(23):4466-79.Page 9 ensure text is the correct size in lines 233-235 –the company names are too large.
Reference list
Ref 2 is this a book? If so the Editors, publishers, City of Publication should be supplied as you have done in Ref 3.
In general the reference list is rather dated with only six references from the last 5 years cited. The suggested references above would bolster useful background information on the structure and function of the basement membrane but the authors will have to extract this information from these references and discuss this information in their revised manuscript.
The authors should also supply a graphical abstract.
Author Response
This is an extremely short manuscript and is probably the shortest publication I have ever been given to review. The authors don't really discuss much of substance in their discussion which is also a reflection on the paucity of data they have generated. The authors are encouraged to expand on their comments in a revised version of the manuscript and to consult with other MDPI publications to see how long these are and the in-depth commentary they cover.
I would have liked to find out what features of type XVIII collagen and perlecan structure were important for the functional properties of basement membrane but the authors did not cover this or an appraisal of what is known in the literature. The literature review is very poor with only a total of 29 publications cited and many are dated. The authors should provide current knowledge on the subject of their manuscript to be worthy of publication. This is potentially achievable with inclusion and expansion of the additional information I have provided.
Response: The authors greatly thank the reviewers for the suggested papers, which have been included in the revised manuscript. A discussion on the findings of this study in the context of the revised literature has also been added in the manuscript, including the important roles of perlecan, and its interactions with type XVIII collagen, in structuring the basement membrane.
General comment
Comment #1:–when citing a publication(s) in square brackets make sure they are within the boundaries of a sentence, then close the sentence with a full stop.
Response #1:The authors thank the detailed observation made by the reviewer, and have edited the reference annotation throughout the manuscript.
Comment #2:Line 58 “Heparan sulfate proteoglycans (HSPGs) are glycoproteins,” - HSPGs are proteoglycans as you have already stated here and not glycoproteins –you need to correct this statement.
Response #2:The authors apologize for the miswriting. “Glycoproteins”was replaced by “proteoglycans”.
Comment #3:All of the figures need some attention. Figure 1 legend needs to be more explanatory. What is the dotted line signifying, what are the yellow arrows labeling. The fluors should be described and the M and E labels explained. The duplicated comment above the legend is not required. Use the legend to Figure 5 as a guide for the amount of information required in a legend. If the comments on lines 93-97 are supposed to be part of the legend they should follow on continuously with the other information. The same applies for lines 114-120 for Figure 2 and similar features in the other figures.
Figure 2, 3, legends need to be more informative. Explain the genes that were measured and what the error bars represent. What was n in this experiment. Explain the statistical comparisons and provide P values. The duplicated comment above the legend is not required. Use the legend to Figure 5 as a guide for the amount of information required in a legend.
Figure 4 legend needs more information. What stain was used in A. Explain annotations of labels used. The duplicated comment above the legend is not required. Use the legend to Figure 5 as a guide for the amount of information required in a legend.
Legend to Figure 5. The duplicated comment above the legend is not required.
Response #3. The authors thank the detailed comment addressed by the reviewer, and apologize for the confusion in the Figure legends during manuscript editing (embedding the figures in the main text). The authors have modified not only the legends to the figures but also the font size and the duplicated comment in each figure.
Comment #4. Table 1 should be provided in the correct format for the journal. Descriptive Title required for the Table.
Response #4:The authors thank the detailed observation made by the reviewer. The table has been updated to the format the journal requests. A descriptive title has also been added.
Comment #5. Only 29 references are cited in this study which is very low, the literature does not appear to have been thoroughly examined.
The following relevant references are missing from the literature cited which does not seem to be very extensive. Ref 5 demonstrates a means of anchoring perlecan to the basement membrane which is relevant to the background of this study and to the functional characteristics of this tissue. The other references below provide important background information for this tissue.
1: Dos Santos M, Michopoulou A, André-Frei V, Boulesteix S, Guicher C, Dayan G, Whitelock J, Damour O, Rousselle P. Perlecan expression influences the keratin 15-positive cell population fate in the epidermis of aging skin. Aging (Albany NY). 2016 Apr;8(4):751-68. doi: 10.18632/aging.100928.
2: Sher I, Zisman-Rozen S, Eliahu L, Whitelock JM, Maas-Szabowski N, Yamada Y, Breitkreutz D, Fusenig NE, Arikawa-Hirasawa E, Iozzo RV, Bergman R, Ron D. Targeting perlecan in human keratinocytes reveals novel roles for perlecan in epidermal formation. J Biol Chem. 2006 Feb 24;281(8):5178-87.
3: Ghiselli G, Eichstetter I, Iozzo RV. A role for the perlecan protein core in the activation of the keratinocyte growth factor receptor. Biochem J. 2001 Oct 1;359(Pt 1):153-63.
4: Knox S, Fosang AJ, Last K, Melrose J, Whitelock J. Perlecan from human epithelial cells is a hybrid heparan/chondroitin/keratan sulfate proteoglycan. FEBS Lett. 2005 Sep 12;579(22):5019-23.
5: Bengtsson E, Mörgelin M, Sasaki T, Timpl R, Heinegård D, Aspberg A. The leucine-rich repeat protein PRELP binds perlecan and collagens and may function as a basement membrane anchor. J Biol Chem. 2002 Apr 26;277(17):15061-8.
6: Gatseva A, Sin YY, Brezzo G, Van Agtmael T. Basement membrane collagens and disease mechanisms. Essays Biochem. 2019 Aug 6. pii: EBC20180071.
7: Sekiguchi R, Yamada KM. Basement Membranes in Development and Disease. Curr Top Dev Biol. 2018;130:143-191.
8: Pozzi A, Yurchenco PD, Iozzo RV. The nature and biology of basement membranes. Matrix Biol. 2017 Jan;57-58:1-11.
9: Halfter W, Oertle P, Monnier CA, Camenzind L, Reyes-Lua M, Hu H, Candiello J, Labilloy A, Balasubramani M, Henrich PB, Plodinec M. New concepts in basement membrane biology. FEBS J. 2015 Dec;282(23):4466-79.
Response #5: The authors greatly thank the reviewer for suggesting these important references, which were included in the revised manuscript. The importance of perlecan in the context of epithelial cell differentiation was described in the Discussion section of the manuscript (pages 207-235).
Comment #6. Page 9 ensure text is the correct size in lines 233-235 –the company names are too large.
Response #6: The authors thank the detailed observation of the reviewer, and have corrected the font size.
Comment #7. Reference list
Ref 2 is this a book? If so the Editors, publishers, City of Publication should be supplied as you have done in Ref 3.
Response #7: Ref. 2 has been updated.
Comment #8. In general the reference list is rather dated with only six references from the last 5 years cited. The suggested references above would bolster useful background information on the structure and function of the basement membrane but the authors will have to extract this information from these references and discuss this information in their revised manuscript.
Response #8: The authors thank again the reviewer for the suggested references, and have discussed the findings of this study in the context of the cited literature.
Comment #9. The authors should also supply a graphical abstract.
Response #9: A graphical abstract has been supplied.
Round 2
Reviewer 2 Report
IJMS 591372-V2
General comments
The revised manuscript is improved. The authors have responded to each of my queries to some degree although the extra discussion of the literature is still somewhat limited despite the author comments to the contrary.
Typos requiring correction.
Line 59 add ‘is’ to the sentence
Line 64 a better word than ‘attains’ should be used here-maybe ‘is substituted with’
Line 112 and 159 mucosal and not mucosa
Line 185 [t26, 31] remove the ‘t’
Line 207, 208 “ --- five distinct domains including an N-terminal domain containing three HS chains. The perlecan core protein ---“ reads better and is more correct English.
Line 214 the term is agrin –correct this typo.
Line 213-215 where is the evidence for this statement
Line 222 ‘the oral mucosa’ better
Line 228 do not start a sentence with ‘And’
Line 228 type XVIII collagen and perlecan have previously been colocalized in BM [Miosge N, Simniok T, Sprysch P, Herken R. The collagen type XVIII endostatin domain is co-localized with perlecan in basement membranes in vivo. J Histochem Cytochem. 2003 Mar;51(3):285-96] but where is the evidence that perlecan and type XVIII interact with each other via the terminal endostatin domain of type XVIII collagen-provide a reference to support this statement or re-word this sentence.
Line 232 Another basement membrane component, type IV collagen reads better
Line 234-236 P should be a capitalized italic type form for P ie P
The manuscript is more complete now with the extra information provided by the authors but is still a rather short manuscript.
Author Response
The revised manuscript is improved. The authors have responded to each of my queries to some degree although the extra discussion of the literature is still somewhat limited despite the author comments to the contrary.
Comment #1 Typos requiring correction.
Line 59 add ‘is’ to the sentence
Line 64 a better word than ‘attains’ should be used here-maybe ‘is substituted with’
Line 112 and 159 mucosal and not mucosa
Line 185 [t26, 31] remove the ‘t’
Line 207, 208 “ --- five distinct domains including an N-terminal domain containing three HS chains. The perlecan core protein ---“ reads better and is more correct English.
Line 222 ‘the oral mucosa’ better
Line 228 do not start a sentence with ‘And’
Line 232 Another basement membrane component, type IV collagen reads better
Line 234-236 P should be a capitalized italic type form for P ie P
Line 214 the term is agrin –correct this typo.
Response #1: The authors thank the detailed observation and positive comment of the reviewer, and have corrected all typographic errors.
Comment #2 Line 213-215 where is the evidence for this statement
Response #2: The authors thank the detailed observation of the reviewer, and have added the following reference for this statement.
Bengtsson, E.; Mörgelin, M.; Sasaki, T.; Timpl, R.; Heinegård, D.; Aspberg, A., The leucine-rich repeat protein PRELP binds perlecan and collagens and may function as a basement membrane anchor. J Biol Chem 2002 Apr 26, 277, (17), 15061-8.
Comment #3 Line 228 type XVIII collagen and perlecan have previously been colocalized in BM [Miosge N, Simniok T, Sprysch P, Herken R. The collagen type XVIII endostatin domain is co-localized with perlecan in basement membranes in vivo. J Histochem Cytochem. 2003 Mar;51(3):285-96] but where is the evidence that perlecan and type XVIII interact with each othervia the terminal endostatin domain of type XVIII collagen-provide a reference to support this statement or re-word this sentence.
Response #3: The authors thank the suggestion made by the reviewer. The C-terminal domain NC1 of type XVIII collagen includes the trimerisation domain (a protease-sensitive region) and the endostatin domain, and is known to strongly interact with perlecan in vitro[Sasaki et al., Structure, function and tissue forms of the C-terminal globular domain of collagen XVIII containing the angiogenesis inhibitor endostatin. EMBO J, 1998].
Additionally, it has been reported that endostatin domain of type XVIII collagen and perlecan are colocalized in BM in vivo[Miosge N et al][54], although it is still unclear whether the full length of type XVIIIcollagenand perlecan can bind to each other in vivo.
The authors have reworded the sentence, and all this information has been described in more details in the manuscript (Lines 235-244).
Comment #4 The manuscript is more complete now with the extra information provided by the authors but is still a rather short manuscript.
Response #4: The authors greatly thank the reviewer for the constructive suggestions. The importance of three distinct variants oftype XVIII collagenwas also described in the Discussion section of the manuscript (Lines 203-217).